# Signal to Sequence Attention-Based Multiple Instance Network for Segmentation Free Inference of RNA Modifications

## Abstract

Direct RNA sequencing technology works by allowing long RNA molecules to pass through tiny pores, generating electrical current, called squiggle, that are interpreted as a series of RNA nucleotides through the use of Deep Learning algorithms. The platform has also facilitated computational detection of RNA modifications via machine learning and statistical approaches as they cause detectable shift in the current generated as the modified nucleotides pass through the pores. Nevertheless, since modifications only occur in a handful of positions along the molecules, existing techniques require segmentation of the long squiggle in order to filter off irrelevant signals and this step produces large computational and storage overhead. Inspired by the recent work in vector similarity search, we introduce a segmentation-free approach by utilizing scaled-dot product attention to perform implicit segmentation and feature extraction of raw signals that correspond to sites of interest. We further demonstrate the feasibility of our approach by achieving significant speedup while maintaining competitive performance in m6A detection against existing state-of-the-art methods.

## 1 Introduction

RNA modifications have been discovered since the 1950s (Cohn & Volkin, 1951; Kemp & Allen, 1958; Davis & Allen, 1957) and have been found to play a prominent role in a wide range of biological processes (Xu et al., 2017; Yankova et al., 2021; Nombela et al., 2021)]. Several methods exist to detect these modifications, most prominently $N^6$-methyladenosine (m6A) (Meyer et al., 2012; Dominissini et al., 2012; Chen et al., 2015; Ke et al., 2015; Molinie et al., 2016; Linder et al., 2015; Koh et al., 2019; Dierks et al., 2021)], pseudouridine ($\psi$) (Schwartz et al., 2014a; Lovejoy et al., 2014; Carlile et al., 2014; Liu et al., 2015), and $N^5$-methylcytosine (m5C) (Squires et al., 2012; Hussain et al., 2013; Huang et al., 2019). These methods, while useful, require specific antibody or chemical reagents as well as experimental expertise that is beyond the reach of most computational labs.

The recent development of direct RNA sequencing technology by Oxford Nanopore (Garalde et al., 2018) allows the direct sequencing of native RNA molecules. The technology works through the use of a motor protein that controls the translocation of RNA molecules through the nanopores, generating an electrical current called squiggle that corresponds to the identity of the molecules passing through the pores Figure 1a. The electrical current is deciphered into a sequence of four RNA nucleotides (G, A, C, U) through a process called basecalling and this involves training either Recurrent Neural Networks (RNN) or Convolutional Neural Networks (CNN) (Boža et al., 2017; Stoiber & Brown, 2017; Teng et al., 2018; Zeng et al., 2020) using Connectionist Temporal Classification (CTC) approach (Graves et al., 2006). The presence of a modified nucleotide often results in a shift in the electrical current which can be exploited for RNA modification detection. Nevertheless, modified nucleotides are rare and so only a short portion of a long RNA squiggle is relevant for modification detection. As a result, segmentation algorithms (Loman et al., 2015; Stoiber et al.) are often used by existing detection methods during preprocessing in order to extract useful signals matching to modified positions (Stoiber et al.; Leger et al., 2019; Lorenz et al., 2020; Ueda; Pratanwanich et al., 2021; Gao et al., 2021; Begik et al., 2021; Parker et al., 2021; Hendra et al.,

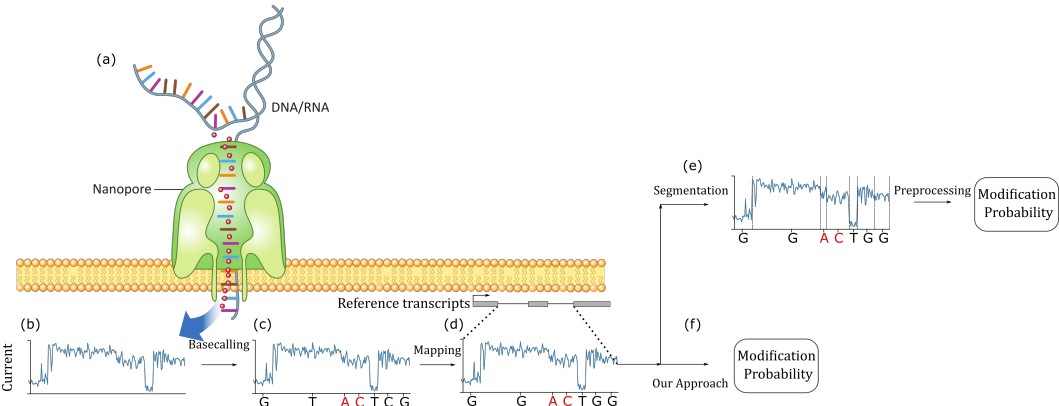

Figure 1: (a) RNA molecule being translocated through the Nanopore. Image is adapted from (Wan et al., 2022)(b) Electrical current, or nanopore squiggle, generated as the RNA nucleotides pass through the Nanopore. (c) The squiggle is deciphered a series of nucleotides through basecalling. RNA modification such as m6A modification can only occur in the presence of AC motif, so signals matching all other motifs might not be useful for detecting m6A modifications.(d) The basecalled sequence is mapped to the reference transcripts, correcting some errors made during basecalling. (e) Segmentation step is performed by most modification detection methods in order to map the squiggle corresponding to the candidate AC motif before further preprocessing and modification prediction. (f) Our method skips the segmentation step and outputs modification probability corresponding to the candidate positions directly

2021; Stephenson et al., 2022; Sethi et al., 2022). However, modifications such as m6A for example, mostly occur within 18 out of the 1024 possible 5-mer motifs (Meyer et al., 2012; Dominissini et al., 2012; Schwartz et al., 2014b) while other modifications such as m5C or pseudouridine only occur within segments containing the C or U nucleotides. Since segmentation algorithms typically segment the entire transcriptome, the modification detection pipeline often requires a huge storage space to store the segmentation results and suffers from slow running time due to the many preprocessing steps required to extract relevant features from the potential modified positions.

In this work we attempt to address these shortcomings by putting together several machine learning techniques that can help to streamline the RNA modification detection process. Firstly, we make use of the deep features learnt by the CTC basecaller, with the aim of integrating modification detection to basecalling process in the future. Secondly, we implement an attention layer between sequence embeddings of the candidate modified positions and the deep CTC features to perform implicit segmentation and feature extraction of the target positions. Finally, to address the issue with noisy modification labels, we implement an end-to-end Attention-based Multiple Instance Learning approach on top of the extracted attention features so as to perform robust classification of modified positions. We validate our approach by performing m6A detection task and demonstrate that our approach is significantly faster than existing m6A detection methods while achieving comparable performance to the current state-of-the-art algorithm. Our work contributes to the field of RNA modification by developing a more scalable solution to RNA modification detection and we hope to drive a wider adoption of machine learning techniques to problems in biology, especially in long read RNA sequencing.

## 2 METHOD

The direct RNA sequencing workflow involves basecalling of RNA squiggles, followed by alignment of the basecalled results to the transcriptome (Figure 1) and for modification detection, another segmentation step is usually required by most detection algorithms. This step is often necessary as RNA squiggle is noisy and modifications only occur on a handful of positions which suggests that most of the signals are not useful for detecting RNA modifications. Nevertheless, segmentation algorithms produce a lot of unused segmented signal regions and most detection algorithms require

further preprocessing steps before predictions can be made, resulting in additional storage overhead and longer compute time. One way to speed up the detection process is to incorporate the modified nucleotides directly to basecalling process but this is significantly more difficult for RNA modification since we usually do not have access to modification labels on a single read level (Linder et al., 2015; Koh et al., 2019). In an attempt to move closer towards streamlining the modification detection process to basecalling, we propose making use of the deep features from the penultimate layer of a RNA basecaller for modification detection. Intuitively, such features contains semantically meaningful representations that correspond to the underlying nucleotides sequence of a given RNA molecule and can be generated during the basecalling process.

Formally, given a signal chunk $\boldsymbol{x} \in \mathbb{R}^L$, and a reference transcript $\boldsymbol{z} = (z_1, \ldots, z_{N_{\boldsymbol{z}}}) \in \{G, A, C, T\}^{N_{\boldsymbol{z}}}$ associated with the signal $\boldsymbol{x} \in \mathbb{R}^L$, our goal is to predict RNA modifications across all possible modified positions in $1 \leq j \leq N_{\boldsymbol{z}}$ within the reference transcript $\boldsymbol{z}$.

To do so, we first learn a *basecaller* that takes $\boldsymbol{x}$ as input and produces a prediction $\widehat{s}(\boldsymbol{x}) \in \{G, A, C, T\}^{|\widehat{s}(\boldsymbol{x})|}$ for the associated underlying sequence. The predicted underlying sequence $\widehat{s}(\boldsymbol{x})$ can subsequently be aligned to the reference transcript $\boldsymbol{z}$. Crucially, we propose to design the basecaller as the composition of two functions:

1. a *feature extractor* $f : \mathbb{R}^L \to \mathbb{R}^{T,D}$ that takes as input a signal chunk $\boldsymbol{x} \in \mathbb{R}^L$ of length $L \geq 1$ and produces a representation vector of fixed length $T \geq 1$ and dimension $D \geq 1$

2. a *decoder* function $g : \mathbb{R}^{T,D} \to \mathbb{R}_+^{T,5}$ that takes as input the representation vector $f(\boldsymbol{x}) \in \mathbb{R}^{T,D}$ is leveraged (using dynamics programming) to produce a prediction $\widehat{s}(\boldsymbol{x}) \in \{G, A, C, T\}^{|\widehat{s}(\boldsymbol{x})|}$ of the underlying nucleotide sequence.

The purpose of learning the basecaller is to identify whether a given signal chunk $\boldsymbol{x}$ contains candidate modified positions. For example, m6A modifications only occur mostly within the DRACH (D=A, G, or U, R=A or G while H is A, C or U) (Meyer et al., 2012; Dominissini et al., 2012), and so the basecalling step is crucial to identify whether such motifs exist within a given signal chunk. On top of identifying the existence of modified motifs within a given signal chunk $\boldsymbol{x}$, we are going to make use of the features $f(\boldsymbol{x})$ for modification detection.

Next, for each of the modified positions $j$ within $\boldsymbol{x}$, we learn a *signal-sequence* feature extractor $h$ that outputs a $D$-dimensional vector $h(f(\boldsymbol{x}), z, j) \in \mathbb{R}^D$ that corresponds to the representation of position $j$ within the transcript $\boldsymbol{z}$ with respect to the deep features $f(\boldsymbol{x})$. This is analogous to performing segmentation and extracting features from segments that correspond to candidate modified positions as described in Figure 1e.

Finally, in order to detect RNA modification at position $j$ of transcript $\boldsymbol{z}$, we collect the set $S_{\boldsymbol{z},j}$ that includes all raw signal chunks $\boldsymbol{x}_i \in \mathbb{R}^L$ harbouring position $j$ of transcript $\boldsymbol{z}$ identified through basecalling and mapping. We extract the representation vectors $h(\boldsymbol{x}_i, \boldsymbol{z}, j) \in \mathbb{R}^M$ for all signal chunks $\boldsymbol{x}_i \in S_{z,j}$ and following earlier work by Hendra et al. (2021), the RNA modification detection problem can then be formulated as a Multiple Instance Learning (MIL) problem (Maron & Lozano-Pérez, 1998) that can be approached with modern statistical learning methods (Ilse et al., 2018; Lee et al., 2019; Cheplygina et al., 2019). The following three subsections describe each step in more details.

## 2.1 BASECALLING

Let $s(\boldsymbol{x}) \in \{G, A, C, T\}^{|s(\boldsymbol{x})|}$ be the true nucleotide sequence associated with raw signal chunk $\boldsymbol{x}$. A perfect basecaller will predict the nucleotide sequence $s(\boldsymbol{x})$ given the raw squiggle chunk $\boldsymbol{x} \in \mathbb{R}^L$. Furthermore, the nucleotide sequence $s(\boldsymbol{x})$ is a continuous subset of the transcript $\boldsymbol{z}$ in a sense that $s(\boldsymbol{x}) = (s_1, \ldots, s_{|s(\boldsymbol{x})|}) = (z_{i(\boldsymbol{x})}, \ldots, z_{i(\boldsymbol{x})+|s(\boldsymbol{x})|})$ for some index $0 \leq i(\boldsymbol{x}) \leq N_{\boldsymbol{z}} - |s(\boldsymbol{x})|$.

Let $f : \mathbb{R}^L \to \mathbb{R}^{T,D}$ be a function parameterized by a neural network (such as CNN or RNN) that transforms the squiggle $\boldsymbol{x} \in \mathbb{R}^L$ to a high dimensional representation vector $f(\boldsymbol{x}) \in \mathbb{R}^{T,D}$ of length $T$ and dimension $D$. Under the CTC approach (Graves et al., 2006), we associate a probability distribution for each of the $T$ high dimensional representation over $\{G, A, C, T, \varepsilon\}$ where G, A, C, T are the four nucleotides and $\varepsilon$ is a *gap character*. The gap character allows the network to produce variable length output since the raw signals can represent variable-length nucleotide sequence

(Silvestre-Ryan & Holmes, 2021). For this purpose, consider a function $g : \mathbb{R}^{T,D} \to \mathbb{R}_+^{T,5}$ that transform the representation vector $f(\boldsymbol{x}) \in \mathbb{R}^{T,D}$ into a sequence of probability vectors. More specifically and slightly abusing notations by identifying $[A, C, G, T, \varepsilon] \cong [1, 2, 3, 4, 5]$, the vector $\pi(\boldsymbol{x}) = g \circ f(\boldsymbol{x}) \in \mathbb{R}_+^{T,5}$ is such that

$$\pi(\boldsymbol{x})[t, A] + \pi(\boldsymbol{x})[t, C] + \pi(\boldsymbol{x})[t, G] + \pi(\boldsymbol{x})(t, A) + \pi(\boldsymbol{x})[t, \varepsilon] = 1$$

for any index $1 \le t \le T$. In other words, at any position $1 \le t \le T$ the vector $\pi(\boldsymbol{x})[t, \cdot] \in \mathbb{R}_+^5$ represents a probability distribution over $\{A, C, G, T, \varepsilon\}$.

The gap characters introduce a many-to-one mapping between the possible path including the gap character and valid nucleotide sequences. For example, the path $G\varepsilon A\varepsilon\varepsilon CC\varepsilon T\varepsilon$ is mapped to the valid nucleotide sequence $GACCT$. We denote this mapping with the operator B in the sense that, for example, $\mathrm{B}(G\varepsilon A\varepsilon\varepsilon CC\varepsilon T\varepsilon) = GACCT$. For a given nucleotide sequence $s \in \{A, C, G, T\}^{|s|}$, we consider the set of sequences $\tilde{s} \in \{A, C, G, T, \varepsilon\}^T$ that maps to $s$ in the sense that $\mathrm{B}(\tilde{s}) = s$. This allows one to define the probability $p(s \mid \boldsymbol{x}) \in (0, 1)$ of the nucleotide sequence $s$ given the raw input $\boldsymbol{x} \in \mathbb{R}^L$ by setting

$$p(s \mid \boldsymbol{x}) = \sum_{\tilde{s}:\mathrm{B}(\tilde{s})=s} \left\{ \prod_{t=1}^{T} \pi(\boldsymbol{x})[t, \tilde{s}_t] \right\}. \tag{1}$$

The predicted nucleotide sequence associated to the raw signal chunk $\boldsymbol{x} \in \mathbb{R}^L$ is obtained by maximizing the score over all possible valid nucleotide sequences $s \in \{A, C, G, T\}^{|s|}$,

$$\widehat{s}(\boldsymbol{x}) = \mathrm{argmax}\big\{ \, p(s \mid \boldsymbol{x}) \; : \; s \text{ is a valid nucleotide sequence} \, \big\}. \tag{2}$$

All these operation can be efficiently approximated with dynamic programming techniques. Since equation 1 describes a likelihood function, the model can be trained by maximum likelihood estimation (Graves et al., 2006). We train a basecaller following this methodology using the open source `Bonito` model (https://github.com/nanoporetech/bonito) with sequence ground truth obtained from `Nanopolish eventalign` (Loman et al., 2015).

## 2.2 EXTRACTION OF SIGNAL-SEQUENCE REPRESENTATION

DNN models are able to learn effective representations that can capture essential aspects of the data domain. Networks trained for object detection, for example, are often trained on image classification tasks for which larger datasets are available Girshick et al. (2014). As a result, features from the deeper layer of the networks tend to be more informative and can easily be adapted to different tasks Mikolov et al. (2013); Donahue et al. (2014); Devlin et al. (2018). Similarly, while RNA modifications are rare, sequence labels for basecalling training are abundant Chen et al. (2021) and should therefore provide rich features for RNA modification detection. Here we view the basecaller training as a form of pretraining task for modification detection where the deep features from the second last layer of the basecaller network can be used to represent a potentially modified nucleotide segments.

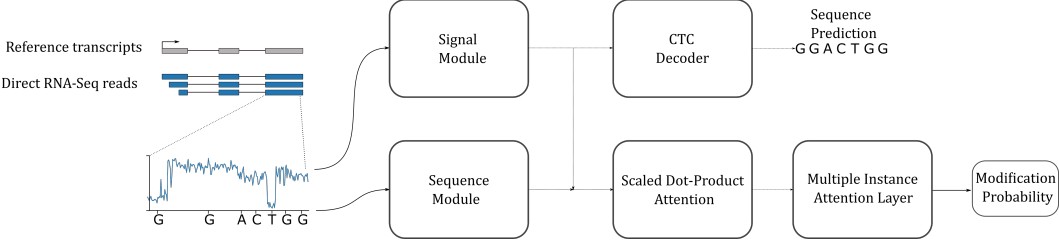

Figure 2: Overview of the model. The signal module $f$ comprises 3 CNN layers followed by 5 LSTM layers while the sequence module $\mathbb{R}$ comprises 3 bidirectional LSTM layers followed by 1 linear layer. The scaled dot product attention combines the output the two modules to produce signal-sequence representation and the CTC decoder $g$ outputs sequence prediction from the signal representation. The MIL attention layer takes in the signal-sequence representation to output modification probability

Given the deep features $f(\boldsymbol{x}) \in \mathbb{R}^{T,D}$, we want to extract the $M$-dimensional signal-sequence representation vector $h(f(\boldsymbol{x}), \boldsymbol{z}, j) \in \mathbb{R}^M$. Our goal here is to predict RNA modification at position $j$ of transcript $\boldsymbol{z}$ and so computing $h$ is analogous to segmenting the $D$-dimensional sequential features of $f(\boldsymbol{x})$ and summarizing its features across a high-dimensional segment that corresponds to position $j$.

To begin, we calculate the embedding of $\boldsymbol{z}$ at position $j$ by passing $K = 10$ flanking nucleotides around the $j$-th position using a bidirectional LSTM. More specifically, given a position $j$ within the reference transcript, consider the sequence $[\boldsymbol{z}]_{j,K} \equiv (z_{j-K}, z_{j-K+1}, \ldots, z_{j+K}) \in \{A, C, G, T\}^{2K+1}$ of $K$-neighbouring nucleotides. The representation of the position $j$ is obtained by passing through the sequence $[\boldsymbol{z}]_{j,K}$ through a bidirectional LSTM denoted as $\mathbb{R} : \{A, C, G, T\}^{2K+1} \to \mathbb{R}^M$ to obtain a representation vector $\mathbb{R}([\boldsymbol{z}]_{j,K}) \in \mathbb{R}^M$.

Afterwards, we compute the representation $h(\boldsymbol{x}, \boldsymbol{z}, j) \in \mathbb{R}^M$ of the $j$-th position within the reference transcript $\boldsymbol{z}$ and whose signal is present within the raw signal chunk $\boldsymbol{x} \in \mathbb{R}^L$ with a standard scaled dot-product attention mechanism (Vaswani et al., 2017).

$$h(\boldsymbol{x}, \boldsymbol{z}, j) = \text{Attention}\Big(\mathbb{R}([\boldsymbol{z}]_{j,K}), [f(\boldsymbol{x})W_K], [f(\boldsymbol{x})W_V]\Big) \in \mathbb{R}^M \qquad (3)$$

for query $\mathbb{R}([\boldsymbol{z}]_{j,K}) \in \mathbb{R}^M$, key $f(\boldsymbol{x})W_K \in \mathbb{R}^{T,M}$ and value $f(\boldsymbol{x})W_V \in \mathbb{R}^{T,M}$. The projection matrices $W_K \in \mathbb{R}^{D,M}$ and $W_V \in \mathbb{R}^{D,M}$ are learnable parameters of the attention mechanism. For query $Q \in \mathbb{R}^M$, key $K \in \mathbb{R}^{T,M}$ and value $V \in \mathbb{R}^{T,M}$, the attention mechanism (Niu et al., 2021) is defined as

$$\text{Attention}(Q, K, V)_m = \sum_{t=1}^{T} \alpha_t V_{t,m} \qquad \text{where} \qquad \alpha = \text{softmax}\left\{\frac{VQ}{\sqrt{M}}\right\} \in \mathbb{R}^T_+ \qquad (4)$$

for any coordinate $1 \le m \le M$. Here, we reason that the signal representation $f(\boldsymbol{x}) \in \mathbb{R}^{T,D}$, where the feature extractor $f : \mathbb{R}^L \to \mathbb{R}^{T,D}$ has been obtained when training the basecaller, can be informative since it is trained to maximize the probability of observing the underlying true nucleotide sequence $s(\boldsymbol{x}) \in \{A, C, G, T\}^{|s(\boldsymbol{x})|}$. Intuitively, the attention concentrates the representation of the signal $f(\boldsymbol{x})$ on a sub-sequence that is the most similar to the positional representation $\mathbb{R}([\boldsymbol{z}]_{j,K})$ of the $j$-th position with the reference transcript $\boldsymbol{z}$.

### 2.2.1 DETECTING M6A MODIFICATIONS

In order to detect m6A modification, we update the parameters of the LSTM-based feature extractor $\mathbb{R}$ while keeping the feature extractor $f$ trained during the basecalling fixed. As empirically demonstrated in Hendra et al. (2021), m6A detection can be formulated as a Multiple Instance Learning (MIL) problem (Maron & Lozano-Pérez, 1998). Recall that $S_{\boldsymbol{z},j}$ is the set of signal chunks $\boldsymbol{x}_i \in \mathbb{R}^L$ that contains some part associated to the position $j$ with the reference transcript $\boldsymbol{z}$. Each raw signal chunk $\boldsymbol{x}_i \in S_{\boldsymbol{z},j}$ can be associated with a binary label $y_{i,j} \in \{0, 1\}$ indicating whether the part of the signal $\boldsymbol{x}_i$ associated to the position $j$ contains m6A modification. However, we do not have access to this label $y_{i,j}$ because of the lack of resolution in the labelling process: instead, we only have access to a single label $y_j \in \{0, 1\}$ representing the modification status at position $j$ of the collective signal chunks. We propose to extract the collection of signal-sequence $M$-dimensional representation vectors $\big\{h(\boldsymbol{x}_i, \boldsymbol{z}, j)\big\}_{\boldsymbol{x}_i \in S_{\boldsymbol{z},j}}$ and pool them following the Attention-based Deep MIL framework of Ilse et al. (2018). Set

$$\text{H}(j, \boldsymbol{z}) = \sum_{\boldsymbol{x}_i \in S_{\boldsymbol{z},j}} a_{\boldsymbol{x}_i} h(\boldsymbol{x}_i, \boldsymbol{z}, j) \in \mathbb{R}^M \qquad (5)$$

where the probability vector $\mathbf{a} = \{a_{\boldsymbol{x}_i}\}_{\boldsymbol{x}_i \in S_{\boldsymbol{z},j}}$ is defined as

$$\mathbf{a} = \text{softmax}\left(\Big\{U \tanh\big(V h(\boldsymbol{x}_i, \boldsymbol{z}, j)\big)\Big\}_{\boldsymbol{x}_i \in S_{\boldsymbol{z},j}}\right) \in \mathbb{R}^{|S_{\boldsymbol{z},j}|}_+. \qquad (6)$$

The matrices $U \in R^{1,H}$ and $V \in \mathbb{R}^{H,M}$ are learnable parameters. The coefficient $a_{\boldsymbol{x}_i} \in (0, 1)$ measures the relative contribution of the raw signal chunk $\boldsymbol{x}_i$ in the collective representation $\text{H}(j, \boldsymbol{z})$. The representation vector $\text{H}(j, \boldsymbol{z}) \in \mathbb{R}^M$ can then be used as a feature vector within a standard logistic regression classifier. The complete model is trained end-to-end by minimizing the cross-entropy loss with stochastic gradient descent.

| Dataset | Mean Acc | Median Acc | IoU |
|---------|----------|------------|-----|
| HCT116 | 91.2% | 93.6% | 88.1% |
| HEK293T | 90.1% | 93.5% | 86.0 |

Table 1: Basecalling Accuracy and Intersection over Union on HCT116 and HEK293T datasets

## 3 EXPERIMENTS

### 3.1 DATASETS

**HCT116 direct RNA Sequencing Data:**   The HCT116 cell line direct RNA sequencing data was provided by the SG-NEX project (Chen et al., 2021). The dataset was spit on the gene level into train, validation, and test sets. We use the training set to train networks for basecalling and m6A detection and use the validation set for model selection. The dataset also comes with m6A labels generated using the m6ACe-seq protocol (Koh et al., 2019). We follow the training procedure of Hendra et al. (2021) where we restrict our training data to sites harbouring DRACH motifs.

**HEK293T direct RNA Sequencing Data:**   The HEK293T cell line direct RNA sequencing data is provided by (Pratanwanich et al., 2021). The m6A labels for this dataset was generated by `m6ACE-seq` (Koh et al., 2019) and `miCLIP` (Linder et al., 2015). We use this dataset to validate both the basecalling performance as well as m6A classification performance. Similar to the HCT116 dataset, we restrict our prediction to the DRACH sites(D=A, G, or U, R=A or G while H is A, C or U).

### 3.2 BASECALLING AND MAPPING ACCURACY

We first evaluate the accuracy of our basecaller and whether it can identify positions within raw signal chunks correctly by aligning the predicted sequence to its underlying reference label using the Smith-Waterman algorithm (Smith et al., 1981). We measure the mapped accuracy as:

$$\text{Accuracy} = \frac{\text{Number of Matched Bases}}{\text{Number of Reference Bases}}. \tag{7}$$

In order to extract accurate feature representation, we need to identify whether the position we wish to model exists within a given read chunk. To do this, we align each predicted sequence $\widehat{s}\boldsymbol{x}_i$ to its reference transcript $\boldsymbol{z}_i$ and obtain a set of predicted positions $\widehat{L}_i$ spanned by read $i$. The alignment step serves to correct small error in the prediction and so we do not necessarily need a very high basecalling accuracy to correctly identify the positions spanned by a given signal chunk. As such, we also measure the Intersection over Union (IoU) of the predicted positions $\widehat{L}_i$ against the ground-truth transcript positions $L_i$ represented in read $i$. This is given by:

$$\text{IoU} = \frac{\widehat{L}_i \cap L_i}{\widehat{L}_i \cup L_i} \tag{8}$$

On the HCT116 cell line, we manage to achieve a mean accuracy of 91.2% and median accuracy of 93.6% while on the HEK293T cell line, it achieves a mean accuracy of 90.1% and median accuracy of 93.5%. Additionally, the model achieves an average IoU of 88.1% on the HCT116 cell line and 86.0% on the HEK293T cell line. This indicates that the model can recognize the underlying sequence of each signal chunk and its mapped alignment coincides strongly with the ground truth label. Another way to improve the mapping quality will be to consider the outputs from two adjacent overlapping signal chunks, a strategy implemented by the original Bonito basecaller (Silvestre-Ryan & Holmes, 2021), which we leave for future work.

### 3.3 COMPARISON OF M6A DETECTION AGAINST EXISTING METHODS

Here we demonstrate the effectiveness of our approach by training our model to perform m6A detection based on the extracted signal-sequence representation. We name our approach `m6Araw` and

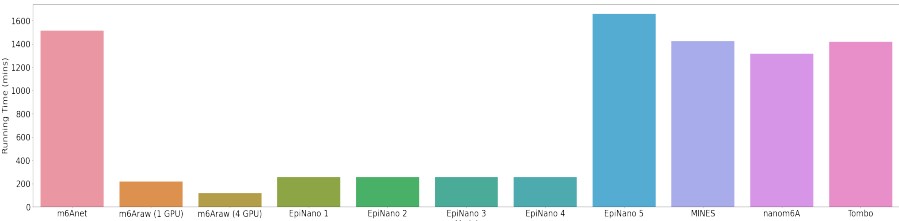

Figure 3: Run time comparison of the models on HCT116 cell line

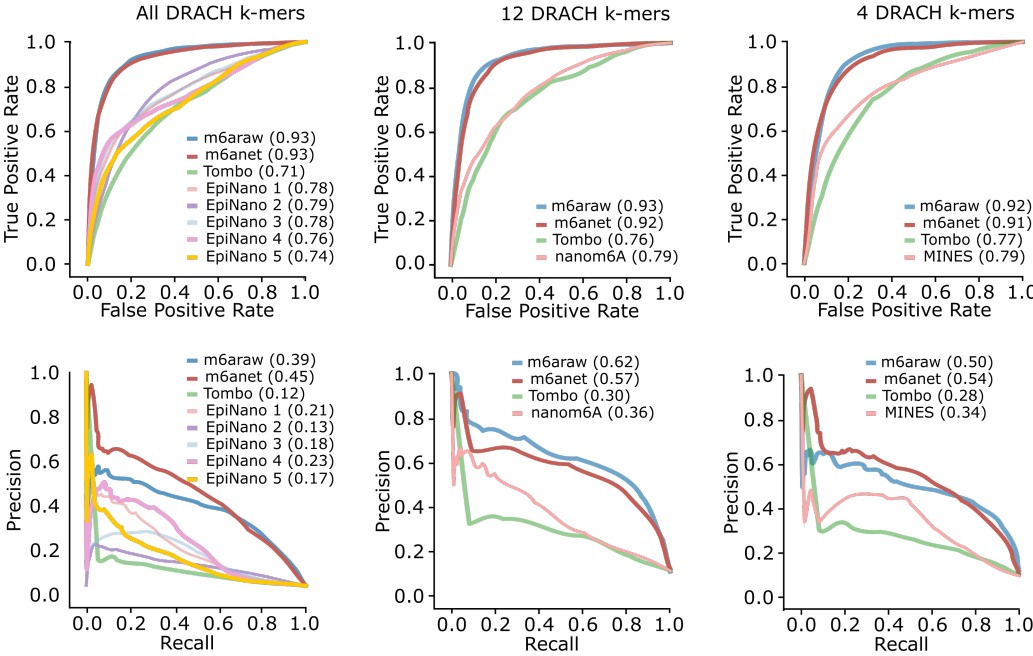

Figure 4: ROC Curves and PR Curves of `m6Araw` and other existing approaches on HCT116 cell line

compare the runtime as well as ROC-AUC and PR-AUC on several partition of the HCT116 against several existing methods (Stoiber et al.; Lorenz et al., 2020; Gao et al., 2021; Liu et al., 2021; Hendra et al., 2021) for m6A detection as detailed in Hendra et al. (2021). Since not all methods detect m6A modifications on all DRACH motifs, we compare `m6Araw` separately with these methods on the subset of the HCT116 test set and the HEK293T dataset that contain motifs required by each approach.

Firstly, we note that m6Araw has the fastest running time out of all other methods on the HCT116 dataset (3) with just one GPU, roughly 6 times faster than m6Anet, the current state-of-the-art method for m6A detection. On the other hand, the EpiNano 1, EpiNano 2, EpiNano 3, and EpiNano 4 models could not be run without crashing our machine with 126 GB of RAM. As such, we have to modify the code for the EpiNAno models discarding all non-DRACH motif motifs during preprocessing step so as to run it successfully. We do not have the run time comparison for the HEK293T cell line since some of the methods require segmentation algorithms that do not work with the HEK293T file format and so additional run time for file conversion is required.

On the HCT116 dataset, `m6Araw` perform comparably (ROC-AUC: 0.930, PR-AUC: 0.385, ROC-AUC: 0.927, PR-AUC:0.609, ROC-AUC:0.917, PR-AUC:0.497) against m6Anet (ROC-AUC: 0.926, PR-AUC:0.451, ROC-AUC: 0.916, PR-AUC:0.565, ROC-AUC:0.908, PR-AUC:0.543) while outperforming all other methods (Figure 4, Table 2). We observe similar results in the HEK293T cell line where our model outperform existing methods (ROC-AUC: 0.818, PR-AUC: 0.319, ROC-AUC: 0.812, PR-AUC:0.333, ROC-AUC:0.796, PR-AUC:0.389) and perform comparably against m6Anet (ROC-

| 5-mer Motifs | Model | ROC-AUC (HCT116) | PR-AUC (HCT116) | ROC-AUC (HEK293T) | PR-AUC (HEK293T) |
|---|---|---|---|---|---|
| 8*18 motifs | `m6Araw` (ours) | **0.930** | 0.385 | 0.818 | 0.319 |
| | m6Anet | 0.926 | **0.451** | **0.838** | **0.366** |
| | Tombo | 0.707 | 0.121 | 0.507 | 0.0857 |
| | EpiNano 1 | 0.776 | 0.206 | 0.710 | 0.240 |
| | EpiNano 2 | 0.788 | 0.133 | 0.725 | 0.182 |
| | EpiNano 3 | 0.781 | 0.176 | 0.722 | 0.213 |
| | EpiNano 4 | 0.764 | 0.235 | 0.704 | 0.227 |
| | EpiNano 5 | 0.736 | 0.167 | 0.670 | 0.170 |
| 4*12 motifs | `m6Araw` (ours) | **0.927** | **0.620** | 0.812 | 0.333 |
| | m6Anet | 0.916 | 0.565 | **0.837** | **0.373** |
| | Tombo | 0.759 | 0.296 | 0.504 | 0.099 |
| | nanom6a | 0.787 | 0.364 | 0.719 | 0.203 |
| 4*4 motifs | `m6Araw` (ours) | **0.917** | 0.504 | 0.796 | 0.389 |
| | m6Anet | 0.908 | **0.543** | **0.825** | **0.440** |
| | Tombo | 0.767 | 0.280 | 0.515 | 0.166 |
| | MINES | 0.792 | 0.340 | 0.708 | 0.326 |

Table 2: Performance comparison of `m6Araw` against existing m6A detection methods

AUC: 0.838, PR-AUC: 0.366, ROC-AUC: 0.837, PR-AUC:0.373, ROC-AUC:0.825, PR-AUC:0.440)
(Figure 5, Table 2). These results suggest that our approach can produce informative signal-sequence
representation to detect m6A modifications and is competitive against existing segmentation-based
approaches.

## 3.4 ROBUSTNESS OF M6A PREDICTION DESPITE LABEL NOISE

The modifications labels used to train and validate `m6Araw` predictions on both HCT116 and
HEK293T datasets often fail to capture genuine modified sites. Different protocols to detect m6A

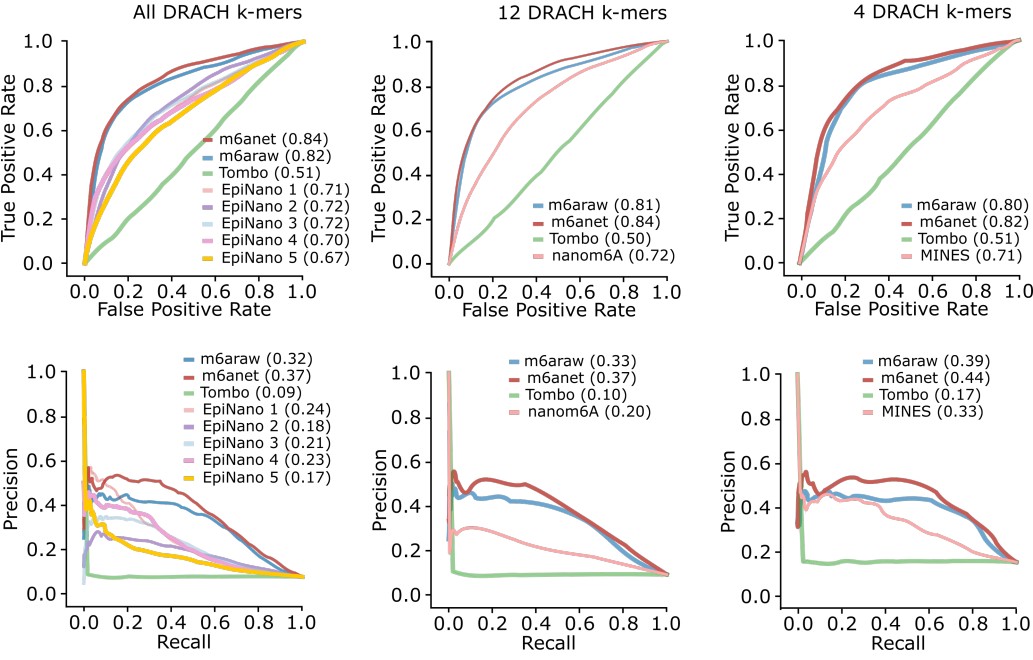

Figure 5: ROC Curves and PR Curves of `m6Araw` and other existing approaches on HEK293T cell
line

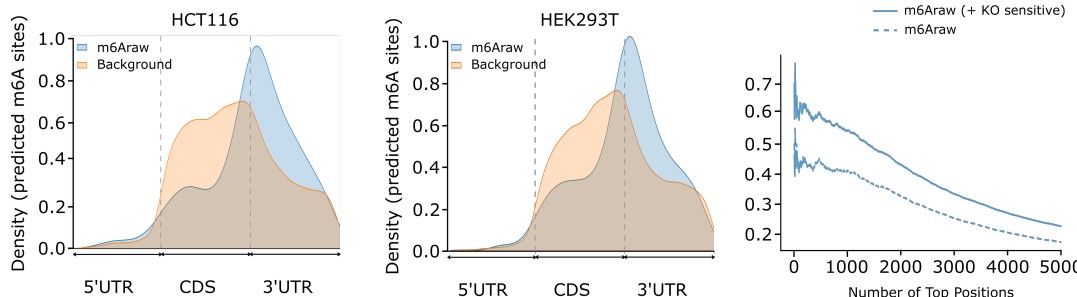

Figure 6: Metagene plots of `m6Araw` prediction on test sets of HCT116 and HEK293T and precision of the top predicted sites for `m6Araw` before and after METTL3-KO

modifications for example have reported different modified sites (Grozhik & Jaffrey, 2018; Koh et al., 2019). This limitation will affect both training and validation of any models trained to detect m6A modifications as the modification labels will be noisy and likely contain a large number of false positives and negatives. Several work has therefore incorporated additional validation criteria based on the known biology of m6A modification. This involves checking for significant deviation in the signal intensity of wild type samples against samples in which METTL3, a known m6A writer, gene has been knocked out (Lorenz et al., 2020; Hendra et al., 2021). Additionally, one can also check for the distribution of m6A predicted sites along each transcript since m6A modification is known to be enriched around the 3'UTR (Linder et al., 2015; Ke et al., 2017)]. Here we reason that our approach displays a robustness to label noise by demonstrating that `m6Araw` can capture sites not previously labelled by miCLIP (Linder et al., 2015) or m6ACe-seq (Koh et al., 2019). We see that `m6Araw` predicted sites display a strong enrichment towards the 3'UTR in both HCT116 and HEK293T cell lines(Figure 6). Furthermore, incorporating METTL3 sensitive sites as modified sites result in roughly 20% increase in the model precision for the top predicted sites, suggesting that our previous precision is underestimated.

## 4 DISCUSSION

In recent years, we have seen an increasing number of computational tools being developed to detect RNA modifications from direct RNA sequencing data. These tools have facilitated a growing number of studies into RNA modifications but at the same time require a lot of compute resources. Our study explores the possibility of streamlining such processes by avoiding extra segmentation and preprocessing steps, and instead, detect RNA modifications directly from raw signals. We demonstrate that our approach can produce informative signal-sequence representations to detect m6A modifications, achieving competitive performance against state-of-the-art method in m6A detection. Furthermore, we have also demonstrated the feasibility of integrating modification detection to basecalling by performing modification detection using features generated by the second last layer of our trained basecaller. In the future we hope to combine RNA basecalling along with detection of other RNA modifications. Lastly, we hope that this work can lay the foundation for further study into representation learning in the context of detecting RNA modifications directly from raw signals.

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

## A    APPENDIX

You may include other additional sections here.

