# OpenReview forum: "Signal to Sequence Attention-Based Multiple Instance Network for Segmentation Free Inference of RNA Modifications"
_ICLR.cc/2023/Conference — Submitted to ICLR 2023_

### Official Review · Reviewer_MYMy · 2022-10-24

**Confidence:** 1
**Correctness:** 3
**Technical Novelty And Significance:** 2
**Empirical Novelty And Significance:** 2
**Recommendation:** 5

**Clarity, Quality, Novelty And Reproducibility:**

The paper was written well and easy to follow, and the algorithms seems not difficult to reproduce (of course further detailed information of algorithms are necessary, e.g., network architectures, optimizers, and detailed training process)
The novelty depends on this new task if nobody has investigated deep learning algorithms for in this field (otherwise, the algorithms and techniques have already been widely used in other field).

**Strength And Weaknesses:**

Strength: While I am not an expert in RNA analysis and detection, the methods used in this study are widely and popularly applied in image and speech processing for recognition and detection task (such as deep feature extraction, CTC sequence mapping and alignment, attention, as well as the multiple instance learning framework), it is a good idea to apply these techniques for the new task. If there is no such kind of research in this specific field (for inference of RND modifications), I believe that it will be a good start to explore these kind of techniques for this new task, and possibly bring a large convenience for advancing the study in this new field.
Weakness: Several information are missed for clearly capture the idea and application: 1. A detailed network architecture framework may be drawn to show the signal flow including feature extraction and position detection (CNN and LSTM, layers, number of neurons, and possible effects of different selection of their size).
2. Training data size, usually in image and speech, the training data size are large, e.g., millions of pictures for images, and thousands of hours of speech. I am not sure for RNA data, how large size of the training sequences. And usually, data of RND modification is rare, how authors deal with this kind of lack of data to balance the training data.


**Summary Of The Paper:**

Authors applied deep learning algorithms to propose a segmentation free algorithm for RNA modification position detection. This study suggested a new tool for analyzing RNA modification position detection with efficient procedures from raw signals.

**Summary Of The Review:**

Authors propose to apply deep learning algorithms (as well as MIL framework) for inference RNA modifications which is segment free from raw signal. The study provided a new efficient way for (RNA modification inference) if there are no or few study in this aspect. The idea seemed work well and could provide a new tool for inference of RNA modification.

---

### Official Review · Reviewer_rbVq · 2022-10-24

**Confidence:** 4
**Correctness:** 3
**Technical Novelty And Significance:** 3
**Empirical Novelty And Significance:** 3
**Recommendation:** 6

**Clarity, Quality, Novelty And Reproducibility:**

Clarity: clean, well written

Quality: good

Novelty:  I am not confident in the novelty because most of the methods are proposed in other areas. For example, CTC is proposed for speech recognition and attentive pooling is used in many places. The contribution would be the whole scheme by combing these methods to achieve the task.

Reproducibility: should be reproduced

**Strength And Weaknesses:**

Strength:

1. The paper seems to be the first to use segmentation-free methods for RNA modifications.
2. Well written with enough experimental comparisons.

Weakness:
1. The authors should mention this is the first one to use segmentation-free methods for RNA modifications somewhere in paper.
2. Typos:
    (1). The formula before eq.1, there are two As, should be T for the second A.
    (2). Section 3.2, s hat x_i, should it be s hat instead?
3. In Section 3.3, could you please add which table you are describing for each paragraph? It would be easier for readers.
4. For the results in Figure 3, I am wondering if you can add the results of testing m6Araw with CPU. It is fair to compare other methods using the same configurations. If the methods are used in practice, CPU time would be a more important factor from my perspective.

**Summary Of The Paper:**

This work proposed a scheme of segmentation-free inference of RNA modifications.

The scheme includes:
1. Basecalling of RNA squiggles: transcribe the current signal into RNA nucleotides (G, A, C, U) using the method of connectionist temporal classification (CTC). The representation of the basecalling, say f(x), will be used for the subsequent basecalling model.
2. Embedding of z at position j: passing K=10 flanking nucleotides around j-th position using bidirectional LSTM.
3. Compute the final representation: using scale-dot attention between the embedding obtained in stage 2 and the basecalling representation f(x).
4. Multiple Instance Learning (MIL): use attentive pooling to aggregate multiple signal chunk representations.
5. RNA modifications detection model training.

Compared to the conventional methods, this work does not require the segmentation process which accelerate the inference.

**Summary Of The Review:**

Since the paper emphasizes the advantage of inference speech for the proposed method because of the segmentation-free process, the CPU time results should be added in Figures. This should be a very important factor either as a fair comparison, or for readers' interest.

Overall, this paper uses many machine-learning techniques in other areas for RNA modifications. The segmentation-free scheme is novel.

This paper is above the acceptance threshold

---

### Official Review · Reviewer_FDNP · 2022-10-24

**Confidence:** 2
**Correctness:** 3
**Technical Novelty And Significance:** 2
**Empirical Novelty And Significance:** 3
**Recommendation:** 3

**Clarity, Quality, Novelty And Reproducibility:**

# Quality

In general, the paper has good quality. The proposed methods are well motivated and correctly applied. While there are many small mistakes, I believe that the paper is generally correct.

The experimentation seems to be performed well. The paper gives multiple comparisons to prior works and provides ROC curves for comparison allowing for deeper analysis.

The quality of citations is low.
 - CTC: Graves et al. paper's contribution was the introduction of CTC and a dynamic programming algorithm for the gradient computation, not the maximum likelihood estimation. This paper also introduced the epsilon, not the cited paper Silvestre-Ryan & Holmes.
- Attention: Vaswani et al., 2017 did not introduce the attention. Niu et al. does not look like a high quality citation.

# Novelty

The main novelty of the paper is the application to the novel task. While this application is clever, the novelty for the machine learning community is low.

# Clarity

I found the introduction to be well written and easy to follow. I am not an expert in biology, but I was able to understand the task and the motivation of the paper.

The Section 2 was much harder to follow. The main reason is that the paper doesn't indicate which methods are existing and which are being proposed. For example, 2.1 describes the standard CTC loss restricted to 5 characters.

Another reason I found it hard to follow is the notation. Specifying the dimensionality is a good practice, but it makes all the expressions much clunkier. Then, using the domain as the argument of a function as in Eq. 3 is a very heavy abuse of notation. VQ in Eq. 4 should be Q K^T. Perhaps, it would be easier to follow if the paper introduced one symbol notation for function outputs. It could also help to give an overview before diving into the details ("we apply an LSTM on features xyz followed by the attention between abc and def").

Fig. 3 and 4 are not readable for the people with problems with color perception.

# Reproducibility

The work should be reproducable.

# Typos

- p.3 "dynamic programming"
- p.3 "dimensional representations"
- p.4 "that transforms the representation"
- p.4 "abusing notation" (and this abuse of notation was not used later in the paper, maybe just remove this phrase?)
- p.4 \pi(t, T)
- p.4 possible paths
- p.4 all these operations
- p.5 QK^T
- p.7 several partitions
- p.9 Several works have


**Strength And Weaknesses:**

# Strengths

The main strength of this paper seems to be the experimental evaluation. While I'm not an expert in the area, I cannot thoroughly review this part. Nevertheless, it seems that the paper conducts a great number experiments with prior and the proposed models on two datasets.

Another strength is how the paper describes the biology part. While my knowledge here is very low, I was able to follow and understand the problem setup.

# Weaknesses

The main weakness is the low novelty of the paper from the point of view of the machine learning community. The methods used in the paper are well known and the main novelty seems to be the application to the new task.

Then, it was hard to follow parts of the Section 2. The description of the ML methods used in the paper is very clunky and contains many small mistakes aggravating the issue.

**Summary Of The Paper:**

The paper works on the problem of RNA sequencing. This problem in the nutshell is to go from the electric signal x to the sequence of the nucleotides z. In particular, the paper considers the sequences that contain some particular RNA modifications. In order to tackle the modification, one needs to adjust the nucleotide sequence by making some changes.

The paper proposes to use several methods: the CTC loss for the alignment of the input to the output; the attention to perform the modification; and the multiple instance learning to train a model for instance modification.

**Summary Of The Review:**

In general I liked the proposed application in this work, but I have to reject it for this conference because the novelty for the machine learning community is low. The secondary reason for rejection is that some crucial parts of the paper are hard to follow.

---

### Official Review · Reviewer_NJ1g · 2022-11-05

**Confidence:** 3
**Correctness:** 3
**Technical Novelty And Significance:** 3
**Empirical Novelty And Significance:** 3
**Recommendation:** 6

**Clarity, Quality, Novelty And Reproducibility:**

See comments above. There is no code so reproducibility can not be assessed and parameters and settings (some are asked about above) can not be figured out either. This will need to be addressed in order to be fully assessed.

**Strength And Weaknesses:**

Overall, we found the paper to be much more clear and better written than a previous version we reviewed for NeuIPS. It was much easier to understand how the model is organized, the motivation behind the modeling approach, and the relation to previous work. The authors also added information about comparative time/memory which is crucial to make a claim about significance.

Admittedly, the proposed method for segmentation free detection of m6A uses sequence to sequence modeling components that by themselves are not novel and commonly applied in other domains (e.g. CTC loss, LSTM, and gap character). Thus, the method itself is not as new as the tailoring of these components to the problem at hand which, as far as we can tell, has not been done. That said, the combination of those components in a sensible way to produce competitive results is not a trivial feat and should be acknowledged, especially with ICLR being focused on learning representations.

Given the above our overall excitement about the paper is not high yet positive. We believe the specific issues listed below are addressable.

1.
Please define the dimensions of the spaces you are dealing with and the relation between them: L, T, D, H. M. What are used sizes of those and why?
2.
Basecalling: Authors report results for this task but not other methods’ performance. We understand it’s not the main topic but it would be good to at least have a sense how well the method does on this task.
3.
This question was asked about the previous version as well but was left unanswered: How does the model perform and generalize to unseen datasets that were not included in training? How does the generalization compare with other existing methods? Mateos et al., 2022 follow Yao et al 2021 and make a difference between classification of read signals from kmers context used for training (but read signals not used) aka “sensor generalization” and ability to classify signals from k-mers contexts not seen during training i.e. k-mer generalization. They also include assessment of effects in cases where multiple (and different) modifications may be present.
4.
What are the motifs in Table 2 left column? How are these params decided upon?
5.
What is “Background” in Fig 6? How are these results supporting your claims exactly?

Minor comments:
The term “deep features” is used repeatedly but not defined.
The paper is peppered with small errors, some of which are listed below. The authors should make a concerted effort to scan and remove those.

P2: The squiggle is deciphered a series of
P3: …. Is leveraged…..
P4: The output the two modules
P6: The dataset was spit on (definitely my favorite)


**Summary Of The Paper:**

In this work the authors present m6araw, a method for detecting m6A modifications directly from raw ONT long reads. As the authors nicely explain, current methods require a two step process: One for translating the raw signal from ONT into RNA sequence, another one for detecting the m6A modifications in those sequence segments. Instead, m6araw performs both tasks simultaneously. One component of the model is trained to do sequence detection while the second part uses the hidden representation from the first part but combines it with additional representation to call the modification sites. The authors evaluate their method on two datasets showing that they perform similar to state of the art methods while enjoying a significant boost in mem/time performance.


**Summary Of The Review:**

Overall the paper offers a method which uses components from other domains, tailored together in a sensible way to address an important question (detection of RNA modifications) with the main advantage being significant improvement in time/memory compared to existing methods.

---

### Decision · Program_Chairs · 2023-01-20

**Decision:**

Reject

**Justification For Why Not Higher Score:**

No rebuttal provided by the authors. Raised concerns regarding the performance of existing techniques, generalization capability, exposition and experimental details are not cleared.

**Justification For Why Not Lower Score:**

N/A

**Metareview: Summary, Strengths And Weaknesses:**

In this paper the authors investigate a segmentation free technique for inference of RNA modifications based on a sequence-to-sequence model that has been widely used in the machine learning community.  The authors show, through experiments on two datasets, that the presented approach can yield competitive performance compared to the SOTA detection methods with significant advantages in inference time and memory requirement.  While the CTC, LSTM and attention mechanism applied in the work have been used in machine learning, it is probably one of the first works to successfully apply the sequence to sequence models in computational biology with good performance. From the perspective of extending it to a new domain, all reviewers consider the novelty stands.  That being said,  concerns regarding the performance of existing techniques, generalization capability, exposition and experimental details are also raised by the reviewers.  Answers to these questions would greatly clarify the concerned issues to gain a better understanding of the work.  Since there is no rebuttal provided by the authors to clear those concerns, the paper can not be accepted given its current form.